# Naturally-Occurring Rare Mutations Cause Mild to Catastrophic Effects in the Multifunctional and Cancer-Associated NQO1 Protein

**DOI:** 10.3390/jpm10040207

**Published:** 2020-11-03

**Authors:** Juan Luis Pacheco-García, Mario Cano-Muñoz, Isabel Sánchez-Ramos, Eduardo Salido, Angel L. Pey

**Affiliations:** 1Departamento de Química Física, Facultad de Ciencias, Universidad de Granada, 18071 Granada, Spain; juanlupacheco@correo.ugr.es (J.L.P.-G.); mariocano@ugr.es (M.C.-M.); isanchezb@correo.ugr.es (I.S.-R.); 2Centre for Biomedical Research on Rare Diseases (CIBERER), Hospital Universitario de Canarias, 38320 Tenerife, Spain; edsalido@gmail.com; 3Departamento de Química Física y Unidad de Excelencia de Química Aplicada a Biomedicina y Medioambiente (UEQ), Facultad de Ciencias, Universidad de Granada, 18071 Granada, Spain

**Keywords:** missense mutation, genetic disease, protein structure-function, genotype–phenotype correlations, multifunctional proteins

## Abstract

The functional and pathological implications of the enormous genetic diversity of the human genome are mostly unknown, primarily due to our unability to predict pathogenicity in a high-throughput manner. In this work, we characterized the phenotypic consequences of eight naturally-occurring missense variants on the multifunctional and disease-associated NQO1 protein using biophysical and structural analyses on several protein traits. Mutations found in both exome-sequencing initiatives and in cancer cell lines cause mild to catastrophic effects on NQO1 stability and function. Importantly, some mutations perturb functional features located structurally far from the mutated site. These effects are well rationalized by considering the nature of the mutation, its location in protein structure and the local stability of its environment. Using a set of 22 experimentally characterized mutations in NQO1, we generated experimental scores for pathogenicity that correlate reasonably well with bioinformatic scores derived from a set of commonly used algorithms, although the latter fail to semiquantitatively predict the phenotypic alterations caused by a significant fraction of mutations individually. These results provide insight into the propagation of mutational effects on multifunctional proteins, the implementation of *in silico* approaches for establishing genotype-phenotype correlations and the molecular determinants underlying loss-of-function in genetic diseases.

## 1. Introduction

Due to technological advances in DNA sequencing technology, we are begining to realize the tremendous genetic variability in the human genome [1,2] and to accurately map the genetic alterations associated with certain diseases [3]. A single human genome carries thousands of missense mutations [1,2,4]. However, the potential physio-pathological implications of this genetic diversity are unclear, partly due to our limited ability to predict the effects of missense variants on predisposition to disease [5,6]. Our limited capacity for establishing large-scale genotype–phenotype correlations can be explained by different reasons: (i) the relationships between molecular, pathogenic and organismal effects are complex, even for diseases with *simple* Mendelian inheritance [1,7,8]; (ii) genotype–phenotype correlations are improved when experimental data for molecular effects of mutations are available, but current predictive tools still underperform experimental characterization, particularly for mild to moderate phenotypes [8,9,10,11]; (iii) although many diseases are caused by loss-of-function mutations [12], an operational definition of *loss-of-function* is difficult because human proteins display many functional and regulatory traits. In this context, genotype–phenotype correlations require an integrated understanding on how a single mutation may affect multiple molecular functions *simultaneously* and, consequently, how these molecular effects translate into pathogenic and fitness consequences [1,13,14,15,16,17,18,19,20,21,22,23].

The molecular mechanisms by which missense mutations cause loss-of-function are many-fold [24] and include accelerated protein degradation [5,9,11,25,26], enhanced protein aggregation [14,15,21], catalytic and regulatory alterations [21,27,28,29] and altered biomacromolecular interactions [10,30]. Importantly, the common molecular origin for all these coexisting mechanisms appears to be the structural and energetic perturbation caused by missense mutations [5,8,11,19,24,26]. These perturbations may differently contribute to the molecular phenotype associated with a given missense variant [19]. Therefore, a fundamental issue to understand the effect of missense variants on multifunctional proteins will come from a deep knowledge on how mutational effects are propagated through the protein structure affecting multiple functional features [19,31].

There are several strategies to develop tools for estimating the impact of missense mutations on protein stability and establishing the *potential pathogenicity* of missense variants that can be broadly classified into: (i) *sequence-based* methods that use multiple sequence analysis combined with simple physical estimations of the mutational effect [5,32]—a clear advantage of these methods is that high-resolution structural information on the protein of interest is not required; (ii) *structure-based* methods that (quantitatively) evaluate mutational effects on protein thermodynamic stability (as a change in the unfolding free energy, ΔG_U_). These are often trained and tested using large datasets of experimentally determined mutational effects on ΔG_U_ [33,34,35,36] and work rather well overall but fail for individual mutations [37]. Importantly, mutational effects are universally distributed across protein structures, with solvent-exposed mutations causing much milder effects than those affecting buried residues [35,36]. These studies have three important implications to understand the functional consequences of the vast genetic diversity present in the human genome. First, between ^1^/_2_ and ^2^/_3_ of all possible mutations have minimal effects on conformational stability [35,36] and mild functional effects in vivo [38]. Second, rather small effects on conformational stability (ΔΔG_U_ ~ −2 kcal·mol^−1^) can subtantially accelerate protein degradation and cause loss-of-function [5,9,11,26]. Third, missense mutations may differently affect *native* vs. *partially folded* states and, consequently, operate through different loss-of-function mechanisms [19,39].

We have recently used the enzyme NAD(P)H quinone oxidoreductase 1 (NQO1; EC 1.6.5.2) to investigate how missense mutations may affect different functional features in a multifunctional human protein [19,39,40,41,42,43,44,45]. NQO1 folds into functional dimers of 62 kDa [46,47] and each monomer has 274 residues divided into two domains: an N-terminal domain (NTD, residues 1–225) that contains a tightly bound FAD and most of the catalytic site and the monomer:monomer interface (MMI), and a C-terminal domain (CTD, residues 225–274) that completes the MMI and the active site [44,46,48,49,50,51]. FAD binding is essential for NQO1 function and intracellular stability, since the wild-type (WT) protein lacking bound cofactor (NQO1_apo_) is highly flexible and rapidly targeted for degradation [41,44,49,52]. NQO1 is a multifunctional protein with enzymatic and non-enzymatic functions [47,53]. NQO1 catalyzes the two electron reduction of quinones, acting in the antioxidant defense, maintaining several vitamers and biomolecules in their reduced state and scavenging superoxide radicals [54,55,56,57,58]. The enzymatic cycle contains two half-reactions following a *ping-pong* mechanism: a first step in which NAD(P)H binds to the FAD-bound enzyme (NQO1_holo_), reducing the flavin to FADH_2_ and releasing NAD(P)^+^, and a second half-reaction in which FADH_2_ reduces the bound substrate regenerating the NQO1_holo_ state [47,49,59]. Dicoumarol (Dic) acts as a competitive inhibitor of NAD(P)H in the first half-reaction and also inhibitis the oxidative (second) half-reaction [53,59,60]. In addition, NQO1 develops key macromolecular interactions with regulatory functions, particularly protein:protein (PPI) and protein:nucleic acid interactions [53,58]. Particularly relevant, NQO1 drives PPI with over 20 partners, including key factors in carcinogenesis such as p53, p73α and HIF-1α and these interactions have chaperoning roles protecting protein partners towards intracellular degradation [53,61,62]. Furthermore, these PPIs are very sensitive to ligand binding to NQO1: reduction of FAD by NAD(P)H generally enhances the interaction, while Dic binding usually prevents it [53,62]. NQO1 seems to play a dual role in different pathological states [47]: first, NQO1 is overexpressed in cancer cells, enhancing the cellular antioxidant capacity. This has prompted research towards inhibiting NQO1 as a cancer therapy (e.g., Dic effectively inhibits cancer cells growth [63,64]). Second, two common single-nucleotide polymorphisms (SNPs, p.P187S and p.R139W) have been found in cancer lines and associated with increased cancer risk. The molecular consequences of these two SNPs have been characterized in some detail. P187S (c.559 C > T/p.P187S, allele frequency of 2.47 · 10^−1^ in human population; gnomAD v.2.1.1.; https://gnomad.broadinstitute.org/gene/ENSG00000181019?dataset=gnomad_r2_1) is strongly associated with cancer development and poor prognosis when it occurs in homocygosis [65]. This variant strongly destabilizes the protein intracellularly due to thermodynamic destabilization of the CTD that enhances its ubiquitination and acts as very efficient site for proteasomal degradation [41,43,44,49,52,66,67,68]. In addition, P187S strongly destabilizes the FAD binding site in the apo-state (NQO1_apo_), decreasing by 10–40 fold the affinity for the flavin cofactor [43,44,45,49]. R139W (c.465 C > T/p.R139W, allele frequency of 3.11 · 10^−2^, gnomAD v.2.1.1) is associated with increased risk of developing acute lymphoblastic leukemia in children [69]. This SNP causes minimal perturbations in the function and stability of the NQO1 protein [40,41,70] but promotes aberrant RNA processing, leading to exon 4 skipping that results in an unstable and inactive NQO1 protein [71]. In addition, the consequences of a rare mutation found in a cancer cell line (COSMIC database, c.718 A > C/p.K240Q) have been characterized in some detail, showing quite mild phenotypic consequences [19,39]. By March 2020, there were 125 missense variants in the gnomAD v.2.1.1 database, whereas in the COSMIC database, there were 41 missense mutations compiled. Interestingly, only five mutations were found in both databases. Therefore, it is striking that so little is known about the functional implications of the genetic diversity of NQO1 in the human population.

In this work, we have investigated the effects of eight rare mutations on the stability and functional features of the NQO1 enzyme using biophysical, structural and bioinformatic approaches. Five mutations were found through high-throughput whole-genome/exome sequencing initiatives (gnomAD) and not particularly associated with cancer development and three are derived from cancer cell lines (COSMIC) (Table 1). These mutations affect residues located in the N-terminal region of NQO1 and show different degrees of evolutionary conservation and expected impact on NQO1 stability and function. L7, T16 and Y20 are *fully* conserved among mammalian species (Figure 1) while V9 and A29 are highly conserved and G3 is not. Actually, the mutation A29T is found in some mammalian sequences while G3 is not even the preferred (*consensus*) residue. We may only consider the mutation V9I as *conservative*, since the rest of mutations will cause large changes in either backbone conformation/flexibility or side-chain size, polarity or charge. Structurally, residues affected by these mutations are located in the most N-terminal loop (L1, Gly3), a fully buried short β-sheet (β1, Leu7 and Val9), a solvent-exposed loop (L2, Thr16) and a long α-helix (α1, solvent-exposed Tyr20 and buried Ala29) (Figure 1A,B). Initial structure-based assessments for their impact on NQO1 stability support that V9I and T16M have neutral effects (ΔΔG < −0.5 kcal·mol^−1^) and G3S, G3D, Y20N and A29T are destabilizing (ΔΔG ~ −1–1.5 kcal·mol^−1^), while L7P and L7R are very destabiling (ΔΔG ≥ −2 kcal·mol^−1^) (note that these algorithms do not consider the effect of bound ligands in the structure).

## 2. Materials and Methods

### 2.1. Protein Expression and Purification

Mutations were introduced by site-directed mutagenesis on wild-type (WT) NQO1 cDNA cloned into the pET-15b vector (pET-15b-NQO1) by GenScript (Leiden, The Netherlands). Codons were optimized for expression in *Escherichia coli* and mutagenesis was confirmed by sequencing the entire cDNA. pET-15b-NQO1 plasmids were used to transform BL21(DE3) cells for protein expression. Typically, 40 mL of Luria–Bertani (LB) medium containing 0.1 mg·mL^−1^ ampicillin was inoculated with transformed cells and grown for 16 h at 37 °C. These cultures were diluted into 800 mL of LB containing 0.1 mg·mL^−1^ ampicillin (Canvax Biotech), grown at 37 °C for 3 h to reach an optical density of about 1.0 and then these were transferred to 25 °C and induced with 0.5 mM isopropyl β-d-1-thiogalactopyranoside (IPTG, Canvax Biotech). After 6 h, cells were harvested by centrifugation and frozen at −80 °C for 16 h. Cells were resuspended in binding buffer, BB (20 mM Na-phosphate, 300 mM NaCl, 50 mM imidazole, pH 7.4) plus 1 mM phenylmethylsulfonyl fluoride (PMSF, Sigma-Aldrich) and sonicated in an ice bath. These *total extracts* were centrifuged (2000× *g*, 30 min, 4 °C) and the supernatants (*soluble extracts*) were loaded into immobilized-metal affinity chromatography (IMAC) columns (Ni-Sepharose, GE Healthcare), washed with 30–40 volumes of BB and eluted in this buffer containing 500 mM imidazole. These eluates were immediately buffer exchanged using PD-10 columns to the storage buffer (50 mM HEPES-KOH pH 7.4) and concentrated to 1 mL (*purified NQO1 proteins*). After centrifugation for 20 min at 21,000× *g* at 4 °C, the amount of protein (in NQO1 subunit) and the FAD content were determined from their UV-visible spectra using: ε_(280)_ = 47,900 M^−1^·cm^−1^ for NQO1 and ε_(450)_ = 11,300 M^−1^·cm^−1^ for FAD [43]. Spectra were collected in a Cary 50 spectrophotometer (Agilent) using 0.3 cm path-length quartz cuvettes. The amount of FAD per NQO1 monomer was determined from the ratio between the absorbance at 450 nm from normalized spectra (in M^−1^·cm^−1^) and the extinction coefficient of FAD (thus assuming that the FAD spectral properties bound to NQO1 resemble those of the free flavin). The lack of significant light scattering (due protein aggregation) was routinely checked. Each NQO1 variant was expressed and analyzed at least three times. The purity of purified NQO1 proteins was checked by SDS-PAGE in 12% acrylamide gels.

The amount of NQO1 in total and soluble extracts was determined by Western-blot. Samples were resolved in 12% acrylamide SDS-PAGE gels and electrotransfered to polyvinylidene difluoride (PVDF) membranes (GE Healthcare) using standard procedures. Immunoblotting was carried out using a primary antibody against NQO1 (F8, sc-393736, at 1:500 dilution) and, as secondary antibody, an anti-mouse IgGκ BP-HRP antibody (sc-516102, at 1:2000 dilution) (both antibodies were purchased from Santa Cruz Biotechnology). Samples were visualized using luminol-based enhanced chemiluminiscence (from BioRad Laboratories). Densitometric analyses were performed using Image Lab (BioRad Laboratories) and ImageJ (http://rsb.info.nih.gov/ij/).

To obtain apo-proteins, FAD was removed by treatment with 2 M KBr and 2 M urea, 1 mM β-mercapto-ethanol in BB and PMSF 1 mM, as previously described [43,44]. Apo-proteins were buffer exchanged to 50 mM HEPES-KOH pH 7.4. All purified proteins were stored at −80 °C upon flash-freezing in liquid nitrogen.

### 2.2. In Vitro Characterization of Purified Proteins

For thermal denaturation experiments, purified NQO1 proteins were prepared at a 2 μM final concentration in monomer in the presence of a 10-fold excess of FAD in 50 mM HEPES-KOH, pH 7.4. Triplicate samples were loaded into 3 × 3 mm path-length quartz cuvettes. Thermal denaturation was carried out in a Cary Eclipse (Varian) spectrofluorimeter equipped with a Peltier element. The instrument was calibrated to correct the operational temperature provided by the instrument and the real one using a thermocouple. Samples were equilibrated at 20 °C for 3 min and temperature was increased up to 70 °C at a 2 °C·min^−1^ scan rate. Fluorescence emission was recorded at 350 nm (slit 10 nm) upon excitation at 280 nm (slit 5 nm). Experimental curves were normalized using pre- and post-transition linear baselines to yield the apparent half-denaturation temperatures (T_m_). For each variant, triplicate measurements were carried out and analyzed using two different protein batches. Thus, data are presented as mean ± s.d. from six experiments for each variant.

For partial proteolysis experiments, thermolysin from *Geobacillus stearothermophilus* (Sigma-Aldrich, Madrid, Spain) was prepared in 50 mM HEPES-KOH, pH 7.4, with 100 mM CaCl_2_. The concentration of thermolysin stock solutions was determined spectrophotometrically using ε_280_ = 66,086 M^−1^·cm^−1^ and small aliquots were stored at ~ 20 μM until used. For experiments, purified NQO1 proteins were prepared at ~10 μM NQO1 with 100 μM FAD in 50 mM HEPES-KOH, pH 7.4 (all concentrations were final concentrations), incubated for 5 min at 25 °C and reactions were initiated by addition of thermolysin (to 0.2–1.2 μM, final concentration of 10 mM CaCl_2_). Aliquots were withdrawn at different times, mixed with 25 mM ethylenediaminetetraacetic acid (EDTA), pH 8, and denatured at 95 °C with Laemmli buffer. Controls without thermolysin were prepared similarly and considered as samples for time zero. Samples were resolved in SDS-PAGE gels (12% acrylamide) and analyzed by densitometry using ImageJ. The decay of the full-length protein was fitted using a single exponential function to provide the first-order kinetic constant *k*_obs_, whereas the second-order rate constant *k_prot_* was determined by calculating the slope of the linear fit of *k*_obs_ vs. thermolysin concentration. Changes in local stability of the primary cleavage site upon mutation ΔΔG_prot(WT-mut)_ were determined by the following equation [73]:$$\Delta \Delta G_{prot\left(WT-mut\right)}\;=\;R\cdot T\cdot ln\;\frac{k_{prot\left(mut\right)}}{k_{prot\left(WT\right)}}$$
where *R* is the ideal gas constant, *T* is the absolute temperature and *k_prot(mut)_* and *k_prot(WT)_* are the second-order rate constants for the mutant and the wild-type protein, respectively. Errors in ΔΔG_prot(WT-mut)_ are those determined by linear propagation of the fitting errors for the *k*_prot_ values.

Fluorescence titrations were carried out at 25 °C using 1 × 0.3 cm path-length cuvettes in a Cary Eclipse spectrofluorimeter (Agilent Technologies, Madrid, Spain). Experiments were carried out in 20 mM K-phosphate, pH 7.4, to decrease the binding affinity of NQO1 for FAD and to allow more accurate characterization of the affinity in tightly binding variants [42]. Then, 20 μL of a 12.5 μM NQO1 stock solution (in subunit) was mixed with 0–500 μL of FAD 10 μM and the corresponding volume of buffer was added to yield a 1 mL final volume. Samples were incubated at 25 °C in the dark for at least 10 min before measurements. Fluorescence spectra were acquired in the 340–360 nm range upon excitation at 280 nm (slits 5 nm), and spectra were averaged over 10 scans registered at a scan rate of 200 nm·min^−1^. Blanks were also measured similarly (containing only buffer) and subtracted. The intensity of the fluorescence at 350 nm (I) vs. total concentration of FAD ([FAD]) was used to determine the apparent dissociation constant *K*_d(FAD)_ using the following equation:$$I\;=\;I_{apo}+\left(I_{holo}-I_{apo}\right)\cdot \left(\frac{\left[NQO1_{apo}\right]+\left[FAD\right]+K_{d\left(FAD\right)}-\sqrt{{\left(\left[NQO1\right]+\left[FAD\right]+K_{d\left(FAD\right)}\right)}^{2}-4\cdot \left[NQO1\right]\cdot \left[FAD\right]}}{2\cdot \left[NQO1\right]}\right)$$
where I_holo_ and I_apo_ are the fluorescence intensity of NQO1_holo_ and NQO1_apo_, respectively, and [NQO1] is the total protein concentration (250 nM). Data from at least two independent titrations using different preparations of apo-proteins for each NQO1 variant were used in the fitting. The apparent change in binding free energy, ΔG_(FAD)_, was determined as follows: ΔG_(FAD)_ = R · T · ln *K*_d(FAD)_. The error associated with ΔG_(FAD)_ was determined by linear propagation from the errors of *K*_d(FAD)_.

Titrations of NQO1_holo_ proteins with dicoumarol (Dic) were carried out by isothermal titration calorimetry (ITC) in an ITC_200_ microcalorimeter (Malvern, Iesmat, Alcobendas, Spain). Purified NQO1 samples were prepared in 50 mM HEPES-KOH, pH 7.4, in the presence of 100 μM FAD and loaded into the calorimetric cell. Dic solutions were prepared from 10 mM stocks (in 100 mM NaOH) and diluted into 50 mM HEPES-KOH pH 7.4 to a final concentration in Dic of 120 μM (and 100 μM FAD) and loaded in the titrating syringe. Experiments were typically carried out by performing an initial injection of 0.5 μL followed by 20–22 injections of 1.75 μL, spaced by 100 s. The instrument operated in the high-feedback mode. Data analysis was carried out upon manual integration of the experimental enthalpograms and fittings were done using a single type of independent binding sites model with the software provided by the manufacturer. Dilution heats were included as a fitting parameter. This analysis yields the number of bindings per NQO1 monomer (N) as well as the association binding constant (*K*_a_, being *K*_a_ = 1/*K*_d_), enthalpy (ΔH) and entropy (ΔS). Experiments were performed at least three times at 25 °C and using two different protein preparations of each variant. Data at this temperature are presented as mean ± s.d. Experiments at 10, 15, 17.5 and 20 °C were performed only once and reported errors were those from fittings. The apparent change in heat capacity (ΔC_p_) was determined from linear dependence of ΔH on temperature.

Structure–energetic relationships were combined with Dic binding calorimetric data to determine the magnitude of the conformational change associated with binding (N_conf_, in number of residues) using previously described procedures [39]. Briefly, two experimental apparent thermodynamic binding variables, ΔH and ΔC_p_, were assumed to be the sum of two separate contributions: one arising from the energetics of rigid-body interactions between *NQO1_holo_* and Dic (*intrinsic binding*) and the other arising from the *conformational change* associated to binding. The former can be estimated from changes in solvent accesible surface upon Dic binding as determined from the crystal structure of the NQO1_holo_ complex with Dic (NQO1_dic_; PDB code 2F1O, [72]), yielding values of −5.8 kcal·mol^−1^ and −0.14 kcal·mol^−1^·K^−1^ for intrinsic ΔH (at 25 °C) and ΔC_p_, respectively. Thus, the contribution from the conformational change (ΔH_conf_ and ΔC_p,conf_) can be obtained from the difference between the experimental variables and those calculated for intrinsic binding. This contribution can be parametrized in terms of the number of residues folded upon binding (i.e., those involved in the conformational change) from well-known structure–energetics relationships for protein folding as follows [39,74]:ΔH_conf_ = 0.215 · N_conf_
ΔC_p,conf_ = 0.0138 · N_conf_
where ΔH_conf_ is given in kcal·mol^−1^ and ΔC_p,conf_ in kcal·mol^−1^·K^−1^. This approach provides two ways of determining N_conf_ from experimental and structural variables.

### 2.3. In Silico Mutagenesis and Structural Analysis

*In silico* mutagenesis of the dimeric NQO1 protein was performed using Rosetta [75]. The atomic coordinate files for the NQO1_holo_ state (PDB code 1D4A) and NQO1_dic_ state (PDB code 2F1O) of the WT protein were processed to remove non-canonical amino acids and water molecules. Subsequently, each PDB file was refined with fast relax constrained to native coordinates using Cartesian-space refinement as described elsewhere [76] and the Rosetta energy function REF15. After relaxation and refinement, the structural model with the lowest energy was used as the starting structure for mutagenesis. For *in silico* mutagenesis, the cartesian version of Rosetta’s ddG protocol [77] was followed and five structures were generated for each variant (WT and eight mutants), both in the NQO1_holo_ and NQO1_dic_ states, following the procedures described in [76]. Finally, the energies (expressed in kcal·mol^−1^) of every structure were calculated and averaged for each variant in a given ligation state. ΔΔG values were calculated by subtracting the average energy of the corresponding mutant ensemble to the average energy of the WT ensemble. As cut-off, ΔΔG values > 0.5 kcal·mol^−1^ were considered as *stabilizing*, ΔΔG values < −0.5 kcal·mol^−1^ were considered as *destabilizing* and ΔΔG values between −0.5 and +0.5 kcal·mol^−1^ were considered as *neutral*. Structural analysis and representations were generated with YASARA molecular modeling software [78].

### 2.4. Structure- and Sequence-Based Analysis of Mutational Effects on Protein Stability and Potential Pathogenicity

To determine changes in protein (thermodynamic stability), we used the following methods: (i) SDM2 (Site Directed Mutator 2, http://marid.bioc.cam.ac.uk/sdm2/prediction) [79] analyses were carried out using chains A and B from three high-resolution crystal structures (PDB codes 2F1O, 1D4A and 5FUQ, [43,72,80]). For each mutant, the value provided was the mean ± s.d. from six analyses; (ii) PoPMuSiC (https://soft.dezyme.com/) [81] analyses were carried out using all chains from three high-resolution crystal structures (PDB codes 2F1O, 1D4A and 5FUQ, [43,72,80]). For each mutant, the value provided was the mean ± s.d. from three analyses; (iii) DynaMut (http://biosig.unimelb.edu.au/dynamut/) [82] analyses were carried out in chains A and C (PDB codes 2F1O and 1D4A, [72,80]) or chains A and B (PDB code 5FUQ, [43]). For each mutant, the value provided was the mean ± s.d. from six analyses; (iv) MAESTRO (https://pbwww.che.sbg.ac.at/) [83,84] analyses were carried out in chains A and C (PDB codes 2F1O and 1D4A, [72,80]) or chains A and B (PDB code 5FUQ, [43]). For each mutant, the value provided was the mean ± s.d. from six analyses; (v) CUPSAT (http://cupsat.tu-bs.de/) [85] analyses were carried in the *Thermal denaturation method* using chains A and B from three high-resolution crystal structures (PDB codes 2F1O, 1D4A and 5FUQ, [43,72,80]). For each mutant, the value provided was the mean ± s.d. from six analyses.

For bioinformatic prediction of pathogenicity, we used: (i) PolyPhen-2 (http://genetics.bwh.harvard.edu/pph2/index.shtml) analyses, which yielded the potential effect of mutations as *Benign*, *Possibly damaging* or *Probably damaging*, using information on multiple sequence alignments and simple physical estimations of mutational effects [32]; (ii) SIFT (https://sift.bii.a-star.edu.sg/) analyses, which used multiple sequence alignment tools to estimate evolutionary tolerance of mutations (*Tolerated* vs. *Not tolerated*) [86]; (iii) PROVEAN (http://provean.jcvi.org) analyses, which used a sequence-based (multiple sequence alignment) approach to predict functional effects of mutations (*Neutral* vs. *Deleterious* effect) [87]; (iv) PON-P2 (http://structure.bmc.lu.se/PON-P2) analyses, which provided mutational effects as *Pathogenic*, *Neutral* or *Unknown* using evolutionary sequence conservation, properties of amino acids and physical and functional annotations of replacements sites [88]; (v) MutationTaster (http://www.mutationtaster.org) analyses, which yielded mutational effects as *Polymorphisms* or *Disease-causing* using DNA sequence alterations and information from human genome variability, evolutionary conservation, splice-site changes, loss of protein features and changes that might affect the amount of mRNA [89]. For methods i–iv, the input was the NQO1 protein sequence, and for method v, the NQO1 cDNA sequence.

## 3. Results and Discussion

### 3.1. Expression Analysis of NQO1 Variants Reveals Dramatic Effects of the Mutations L7R and L7P on Protein Stability and/or Solubility

To characterize the effects of rare NQO1 mutations on the stability and functional features of this protein, we expressed them in *E. coli* and purified the WT and mutant proteins. Two of the mutants (L7R and L7P) showed largely reduced expression levels as soluble protein and were not amenable for purification, supporting that these two mutations severely perturb NQO1 folding and prevent the formation of stably folded dimers (Figure 2A,B). Accordingly, structure-based analysis of protein destabilization supported that these two mutations should be the most deleterious (Table 1). Previous expression analyses have also shown similar behavior for largely disrupting mutations at the P187 site (a residue buried in the structure and close to the MMI), such as P187R, P187E and P187L [19,39]. The remaining six mutants were amenable for purification and showed comparable yields to those of WT NQO1 (Figure 2B,C).

### 3.2. Thermal Stability Analyses Revealed Significant Perturbation of the MMI by Mutations T16M and Y20N

For the six mutants amenable for purification, we analyzed and compared their stability with that of the WT protein in the presence of an excess of FAD (NQO1_holo_) and using fluorescence-monitored thermal denaturation (Figure 3). To some extent, the thermal stability of NQO1 variants allows to capture the perturbation caused by the mutations on the MMI since the dimer dissociates prior to the rate-limiting step of the irreversible denaturation [15,39,41]. T16 and Y20 are within or in very close proximity to the MMI and mutations at these residues are potentially the most damaging for dimer stability (Figure 3A). Consequently, the mutations T16M and Y20N, which may introduce mild to moderate structural perturbations (Table 1), led to significant changes in thermal stability (i.e., their T_m_ were lower than that of the WT protein by 4–5 °C) (Figure 3B,C). L7 and V9 are at the shortest distance of 8 Å from the MMI (Figure 3A). Accordingly, the severely perturbing L7P and L7R mutation may significantly destabilize the MMI, thus explaining the dramatic effects of these mutations that prevent dimer formation or severely affect its stability (Figure 2). The mild effects of the mutation V9I likely reflect a small structural perturbation in some proximity to the MMI (reducing thermal stability by 1.5 °C; Figure 3B,C). G3 and A29 are over 15 Å away from the MMI (Figure 3A), and consequently, the effects of the mutations G3S, G3D and A29T in thermal stability were marginal (Figure 3B,C).

It is important to note that the above-mentioned interpretation of mutational effects on NQO1 thermal stability simply implies that both the magnitude of the perturbation caused by the mutation and its distance-dependent propagation to the MMI are key features. Previous systematic mutational studies at the P187 and K240 residues of NQO1 generally supported this interpretation [39]. In the next sections, we will apply a similar approach to other features of NQO1, such as the mutational effects on local stability and binding of functional ligands.

### 3.3. The Local Stability of the Thermolysin Cleavage Site (TCS) Is Reduced by the Distant Mutations T16M and Y20N

Limited proteolysis of NQO1 is a sensitive method to probe the local stability at different regions of the protein [41,50]. In particular, proteolysis by thermolysin typically provides information on the local conformational stability of the region surrounding the primary cleavage site for this protease (i.e., the thermolysin cleavage site (TCS)) located in the NTD of NQO1_holo_; it cleaves between S72-V73) [41]. For instance, the phosphomimetic mutation S82D locally destabilizes this region, accelerating 30-fold proteolysis by thermolysin of NQO1_holo_ (i.e., it causes 2.0 kcal·mol^−1^ of local destabilization in terms of ΔG_prot_ as the mutational effect on the unfolding free energy between the native and cleavable states) [91]. It must be noted that all mutations studied in this work are located at least 15–20 Å from the TCS (Figure 4A).

Proteolytic patterns of all mutant NQO1_holo_ proteins studied resembled that of WT NQO1 (Appendix A), supporting initial cleavage between S72-V73. Furthermore, proteolysis rate constants (*k*_obs_) for all NQO1 variants showed a linear dependence on protease concentrations (Figure 4B and Appendix A), thus implying that changes on the second-order rate constant *k*_prot_ are related to effects on the thermodynamic stability between the native and cleavable state in the TCS (Figure 4C,D) [41]. We found that only the mutations T16M and Y20N caused local destabilization beyond the experimental error (by ~1.1 and 0.4 kcal·mol^−1^, respectively; Figure 4D and Appendix A). Thus, the structural perturbation introduced by these two mutations is *sensed* by the TCS, located 15–20Å away (Figure 4A).

### 3.4. The Mutations T16M, Y20N and A29T Perturb FAD Binding

The binding of FAD to NQO1 is very sensitive to structural perturbations caused by mutations in the NQO1_apo_ and NQO1_holo_ states [39,40,41,42,43,44,45,91]. Actually, when a single mutation reduces significantly FAD binding affinity (by at least 5-fold, i.e., about 1 kcal·mol^−1^ in binding free energy), this often decreases the FAD content of the protein upon purification from *E. coli* cell cultures [39,40,43,44,91]. Since bound FAD is mandatory for NQO1 catalytic function, these effects would affect the specific activity of the enzyme. Structurally, residues T16 and Y20 are close to the FAD molecule (≤5 Å), whereas V9 and A29 are at 12–13 Å and G3 is more than 25 Å away (Figure 5A).

The FAD content in purified NQO1 proteins was determined using near-UV/visible absorption spectroscopy. All the mutants showed high levels of FAD bound, with fractions in the range 0.8–0.9 (similar to those of the WT protein), with the only exception of T16M that showed reduced content (0.5 mol FAD/NQO1 monomer, Figure 5B,C). To evaluate quantitatively mutational effects on FAD binding affinity, we carried out titrations of apo-proteins with FAD monitored by fluorescence spectroscopy (Figure 5D,E, Appendix A and Appendix A). As expected from its lower content in FAD as-purified, the mutant T16M showed a 12-fold lower affinity that corresponds to a decrease in binding free energy of about 1.5 kcal·mol^−1^. In addition, the mutants Y20N and A29T showed decreased binding affinity (3- and 6-fold lower than that of WT NQO1, respectively, that correspond to changes in binding free energy of 0.7 and 1.0 kcal·mol^−1^, respectively) (Figure 5C–E and Appendix A). It is interesting to note the effect of the mutation A29T, which involves a residue located at more than 13 Å away from the FAD binding site.

### 3.5. None of the Mutations Affect Dic Binding Affinity or Energetics

Dic strongly inhibits NQO1 with a *K*_d_ of 10–50 nM, acting as a competitive inhibitor of the NADP(H) coenzyme and the substrate [43,48,63,72,92]. Investigating the affinity and energetics of Dic binding to NQO1 mutants may provide insight into mutational effects on several protein traits: (i) Dic (and analogues thereof) inhibits the growth of cancer cell lines, presumably by preventing the high antioxidant activity provided by cancer-associated overexpression of NQO1 [63,93,94]. Consequently, alterations of Dic binding due to missense mutations may yield decreased cellular sensitivity towards this potential pharmacological treatment for cancer; (ii) Missense mutations can decrease the binding affinity for Dic by altering the conformation and energetics of the native state, shifting the conformational equilibrium towards non-competent states for binding. An excellent example of this behavior is observed in the common polymorphic variant P187S that binds Dic with about a 10-fold lower affinity than the WT protein and shows clearly different binding energetics associated with the folding of the CTD upon Dic binding [39,44,92]; (iii) Extensive mutational studies at the P187 and K240 sites have shown that long-range communication (over 15 Å from the binding site) of mutational effects acting on the conformation and stability of the CTD can be accurately characterized by a detailed analysis of Dic binding energetics [19,39]. As we show in Figure 6A, all the mutations investigated in this work are structurally located far from the Dic molecule (by at least 15 Å).

We have analyzed Dic-binding affinity and energetics by ITC (Figure 6 and Appendix A). All the NQO1 mutants bound Dic with an affinity comparable to that of NQO1 WT and close to the technical limit of the technique. The largest differences correspond to a mild 2-fold change, which implies changes in binding free energy ≤ 0.5 kcal·mol^−1^ (Figure 6B,C and Appendix A). The apparent enthalpic and entropic contributions to binding were also very similar (within the experimental error, see Appendix A and Appendix A). In addition, two apparent thermodynamic binding parameters (ΔH and ΔC_p_) can be used to evaluate the magnitude of the conformational change associated with Dic binding [39,44] (Appendix A and Appendix A). Basically, these two parameters are related with the sum of an intrinsic binding contribution (that can be estimated from the crystal structure of NQO1_dic_) and a second term originating from the conformational change induced upon ligand binding [39]. This second term is straighforwardly converted to the *number of residues involved in the conformational change* using structure–energetic relationships for protein folding (thus yielding the parameter N_conf_). Dic binding caused a minimal conformational change upon binding to all variants, with N_conf_ values (average of the two methods) of 10–20 residues (Figure 6D and Appendix A).

### 3.6. Structural Analysis and Energy Calculations Provide Insight into Mutational Effects on Protein Stability and Function

To provide further molecular insight into the effects of this set of eight mutations on different NQO1 traits, we carried out in silico mutagenesis and structural and energetic analysis in both the holo-forms (NQO1_holo_) and Dic bound forms (NQO1_dic_). An overview of the most relevant structural consequences for each mutation are compiled in Figure 7.

#### 3.6.1. The Mutations G3S and G3D

Our experimental analyses have shown that the mutations G3S and G3D are essentially neutral. In the G3S mutant, modeling shows that S3 is surrounded by an hydrophobic patch formed by W35 and W216 and by two negatively charged amino acids, namely D96 and E213, in both NQO1_holo_ and NQO1_dic_ forms (Figure 7A,B). The mutation G3S promotes local changes in its neighboring amino acids but also displaces the N-terminus of the protein towards the opposite direction of its side-chain, thus helping accommodate the perturbation generated upon the introduction of a bulkier and polar residue. Overall energetic scoring by Rosetta suggested that this mutation has a mild–moderate destabilizing effect on NQO1_holo_, whereas this effect is mild in the NQO1_dic_ state (Table 2). The most relevant contributions to this destabilization in both the NQO1_holo_ and NQO1_dic_ states seem to arise from (Appendix A): (i) unfavorable repulsive energy (Δfa_rep), whose increase accounts for an increment in the number of atoms that come into contact upon mutation, which is in agreement with the mutated residue bumping into other residues, such as W35 and W216, thus affecting the repulsive van der Waals energy; (ii) unfavorable isotropic solvation energy (Δfa_sol), which arises from burial of polar residues upon mutation (i.e., primarily that of the side chain of S3 that becomes embedded in the hydrophobic patch located around the mutation site).

Modeling of the mutation G3D shows larger structural and energetic distortions than those observed for the mutation G3S (Figure 7C,D), predicting moderate–large destabilizing effects in the NQO1_holo_ and NQO1_dic_ states (Table 2). This difference between the mutations G3S and G3D can be easily explained due to nature of the mutations, since G3D introduces a bulkier and charged residue that perturbs the surrounding hydrophobic patch, particularly residues W35 and W216—especially the latter one, which undergoes a 100° rotation around its χ_1_ angle in the NQO1_dic_ state and 169° around χ_2_ in the NQO1_holo_ state. This local conformational change generates a significant penalization due to the increase in the repulsive van der Waals energy and unfavorable isotropic solvation energy terms, presumably due to the insertion of a negatively charged residue into the hydrophobic patch formed by W35 and W216 (an effect seen in both NQO1_holo_ and NQO1_dic_ forms) (Figure 7A–D and Appendix A). This may explain why the N-terminus is pushed a few Å away from the mutated site (Figure 7C,D, green arrow shows direction of displacement) in the model of the G3D mutant. This could suggest that the N-terminus of the protein might help to reduce and prevent clashes by potentially accommodating some of the observed perturbations. This also possibly explains the minimal changes in thermal stability and ligand binding observed experimentally, although the mutation is perceived as *destabilizing* by Rosetta.

#### 3.6.2. The Mutations L7P and L7R

Our experimental analyses have indicated that the mutations L7P and L7R are heavily destabilizing (Figure 2). Regarding the mutation L7P, we must note that proline is rarely found in the middle of β-sheets as its presence is disfavored because of the lack of accessible backbone N-H group to participate in hydrogen bonding, thus disrupting the β-sheet structure [95,96]. Energetic scoring by Rosetta analysis supports the tremendous destabilizing effect of the mutation L7P in both NQO1_holo_ and NQO1_dic_ states (Table 2). L7 is surrounded by a hydrophobic environment in the core of the protein, composed by A6, I8, V9, V38, L92, V98, F100 and W116 (Figure 7E,F). Most of the hydrophobic interactions established by L7 in this pocket are essentially abolished in the L7P mutant (in both NQO1_holo_ and NQO1_dic_ states), likely due to the cavity created by the mutation (Figure 7E,F). In the WT protein, there is only a small cavity with a volume of 17 and 31 Å^3^ (NQO1_holo_ and NQO1_dic_ forms, respectively), whereas the L7P mutation increases the size of this cavity to a volume of 71 and 109 Å^3^ (NQO1_holo_ and NQO1_dic_ forms, respectively). This cavity in the protein core may be, in part, responsible for the great destabilizing character of the mutation, as reflected in the energetic penalization in the attractive energy (fa_atr) term (Appendix A). Interestingly, parametrizations of the conformational destabilization caused by cavities in model proteins have provided values in the range of 24–36 cal·mol^−1^·Å^3^ [97,98], thus allowing to calculate that just this cavity (without considering additional perturbations caused) would destabilize the native structure by 1.3–1.9 kcal·mol^−1^ (NQO1_holo_) and 1.9–2.8 kcal·mol^−1^ (NQO1_dic_). Additionally, in both NQO1_holo_ and NQO1_dic_ states, the mutation L7P breaks a backbone hydrogen bond between β1 and β3 parallel strands that destabilizes the β-sheet structure (Figure 8B,E), thus reducing the long-range hydrogen bonds energy term (hbond_lr_bb) (Appendix A). The perturbation generated by P7 is accentuated by the distortion in dihedral Φ and Ψ angles caused by the geometry of the mutated residue and the prevention of hydrogen bond formation in the proximity of the mutated site (Figure 8C,F).

The mutation L7R is also highly destabilizing, as calculated by Rosetta (Table 2). This mutation introduces a severely perturbing, positively charged residue in the formerly described well-packed hydrophobic pocket (Figure 7G,H). Unlike L7P, L7R does not generate a cavity (Figure 7E,F) but instead causes steric clashes with neighbouring residues (Figure 7G,H and Appendix A) in both NQO1_holo_ and NQO1_dic_ states. In this case, conformational destabilization mainly arises from unfavorable isotropic solvation energy (Δfa_sol) (Appendix A), likely due to the burial of a charged residue in a hydrophobic environment.

#### 3.6.3. The Mutation V9I

Experimental characterization of the mutation V9I has shown nearly neutral effects. As analyzed by Rosetta, this mutation has marginal effects on conformational stability (Table 2). V9 (in NQO1 WT) and I9 (in the V9I mutant) are embedded in a similar hydrophobic environment to that of L7 (that is perturbed by the mutations L7P and L7R) (Figure 7I,J). Interestingly, the mutation V9I causes a favorable change in the van der Waals energy (fa_atr) term (Appendix A) indicating increased van der Waals interactions with neighbouring hydrophobic residues (Figure 7I,J and Appendix A) in both NQO1_holo_ and NQO1_dic_ states. Even though the introduction of a larger amino acid causes some structural strain (i.e., slightly displaces some neighbouring amino acids), this effect is essentially canceled out by the gain in favorable van der Waals interactions (Appendix A).

#### 3.6.4. The Mutation T16M

Our experimental analyses have shown that T16M has phenotypic consequences in several traits, particularly FAD binding, thermal stability and local stability of the TCS. Analysis by Rosetta shows that T16M mutation should have small effects on conformational stability (Table 2). Interestingly, decomposition of energetic contributions provided by Rosetta (Appendix A) indicates different and opposing energetic contributions that nearly cancel out. A detailed structural analysis may provide further insight into these effects as well as in the phenotypic consequences of this mutation (Figure 7K,L).

The mutation T16M produces mild perturbations of the NQO1_holo_ state (Figure 7K), causing small side-chain adjustments affecting residues R15, Y20, E24 and N65. Y20 and E24 residues are located in helix α1 and their rearrangement, together with the presence of M16, produces a small displacement in the FAD binding pose that could explain the effect of T16M on FAD binding.

Interestingly, we also observed noticeable changes in the modeled structures of NQO1_dic_. The residue M16 (located in the first shell of interaction with the ligand FAD at a shortest distance < 5 Å) interacts through hydrophobic contacts with the residues nearby, such as the aliphatic portion of E14 side chain, and brings much closer the residue Q67. This, together with the perturbation in the backbone atoms of the residues adjacent to the mutation site, generates a moderate change in conformation on some residues close to the ligand binding site, ultimately altering its initial pose (i.e., that present in the WT). The residues affected by direct or indirect interaction with M16 are H12, E14, M16 (backbone) and F18, but the amino acids suffering the largest conformational changes on their side chains are Q67 and R15 (100° and 61° rotation across χ_2_ and χ_3_ angles, respectively), whose motions appear to distort important interactions for ligand binding (see Figure 7L and Appendix A). The most significant change results in the reorientation of Q67, which now faces M16 side chain and, while it maintains its original hydrogen bond with FAD, forms another hydrogen bond with the carbonyl group present in F66 residue. The other large conformational change involves R15, which was engaged (in the WT NQO1_dic_ state) in a hydrogen bond network with residues Y20 and E24 (located in helix α1). The mutation T16M disrupts the hydrogen bond with Y20 and makes R15 form two hydrogen bonds with D41 (achieving a virtually similar conformation to that shown in the WT NQO1_holo_ state, see Appendix A). It is interesting to speculate that since Dic is a competitive inhibitor of both the NAD(P)H and the substrate, these structural alterations may have consequences on the NQO1 catalytic cycle.

The residue M16 (as well as the wild-type residue T16) is part of the MMI. In fact, the changes described above can also be translated to some alterations of the MMI (Figure 8G–J) and may explain the effects of T16M in both thermal stability and FAD binding. The adjustments and side-chain reorientations described earlier are summarized in Appendix A (morphing conformations between the WT and T16M as NQO1_dic_) and should be compared to Appendix A (morphing conformations between WT NQO1_dic_ and NQO1_holo_ states of the protein).

#### 3.6.5. The Mutation Y20N

The mutant Y20N shows noticeable effects on thermal stability and FAD binding. Stability calculations using Rosetta provide moderate destabilizing effects (Table 2). Inspection of the modeled structures revealed certain structural alterations caused by this mutation (Figure 7M,N). Introduction of the smaller N20 residue prevents the formation of hydrogen bonds (in both NQO1_holo_ and NQO1_dic_ states) observed between Y20 and residues E24 and R15 (note that the latter residues also played important roles in the structural alterations caused by the mutation T16M). This alteration in hydrogen bonding causes slight backbone and side-chain movements on the residues nearby that are also part of the α1 helix, which also played a noteworthy role in the alterations experienced upon T16M mutation. These adjustments in the NQO1_dic_ state cause slight changes in FAD binding pose (Appendix A), comparable to those observed for the T16M mutation. In the holo form of the protein, a similar situation occurs when Y20 residue is mutated to N20. The broken hydrogen bond network destabilizes the surroundings of the mutated site, ultimately altering ligand binding pose (Appendix A). In this case, E24 also rotates, by ~92°, its side chain around the χ_2_ angle, presumably due to the inability to establish a hydrogen bond with residue N20. Overall, these results provide structural insight into the effect of Y20N on FAD binding affinity and also predict that this mutation might also affect the catalytic cycle of NQO1.

#### 3.6.6. The Mutation A29T

The mutation A29T has essentially neutral effects, except for a moderate decrease in FAD binding affinity. Stability calculations using Rossetta suggest that this mutation substantially destabilizes the NQO1_holo_ state (Table 2). Structural modeling shows that the T29 in this mutant establishes an intra-helix hydrogen bond between its side chain hydroxyl group and the carbonyl backbone group of amino acid A25 in both NQO1_holo_ and NQO1_dic_ states (Figure 7O,P), which might stabilize helix α1. In the NQO1_holo_ form, this mutation causes side-chain conformational regroupings in the vicinity of the mutated site. However, in the NQO1_dic_ state, these local adjustments propagate through helices α1 and α9 towards the FAD binding site and, ultimately, slightly modify its binding mode (See Appendix A) by forming a hydrogen bond with Oγ1 from residue T148 in the FAD binding site (Figure 8K). This interaction might contribute to the destabilizing effect of this mutation when the inhibitor is bound (as calculated by Rosetta) (Table 2).

### 3.7. The Role of Protein Local Dynamics and Stability on Mutational Effects

Protein structural dynamics are, likely, critical to understand many features of NQO1, such as the stabilizing effect of FAD binding towards proteasomal degradation of WT NQO1, ligand binding-mediated changes in the interaction with other proteins and the functional and stability alterations caused by missense mutations and polymorphisms [19,39,41,52,53,90,91]. Recently, we used hydrogen/deuterium exchange (HDX) monitored by mass spectrometry to provide a high-resolution map of the changes in protein local dynamics and stability of WT NQO1 in different ligation states (NQO1_apo_, NQO1_holo_ and NQO1_dic_) [90]. This study has allowed identification of a stable core (as non-exchanging segments) in NQO1_apo_ that holds the NQO1 dimer, as well as the short- and long-range propagation of ligand binding effects (i.e., FAD and Dic). We have, thus, used this dynamic information in different ligation states to provide further molecular insight into the mutational effects in different protein traits experimentally characterized in this work.

Residues L7 and V9 belong to regions with very high structural stability in the NQO1_apo_ state (i.e., these are located in the stable core) (Figure 9 and Appendix A and Appendix A). It is plausible that a significantly large structural perturbation at these sites would have dramatic effects on the ability of NQO1 to fold or the stability of the folded protein. Consequently, we could not obtain stable and soluble NQO1 protein with the largely-disrupting L7P or L7R mutations (Figure 2), whereas the more conservative V9I mutation mildly decreased dimer stability (Figure 3). Residues G3 and A29 are located in regions with moderate structural stability and their stability is hardly sensitive to ligand binding (Figure 9 and Appendix A, Appendix A). Accordingly, mutations G3S, G3D and A29T have mild effects on protein stability (Figure 3) and ligand binding, except for the six-fold decrease in FAD binding affinity in A29T (Figure 6). Residues T16 and Y20 are found in regions with low structural stability in NQO1_apo_, although these regions manifest large structural stabilization by the stepwise binding of FAD and Dic (Figure 9 and Appendix A, Appendix A). Therefore, these analyses help to explain that mutations T16M and Y20N have moderate effects on protein stability (Figure 3) but exhibit important alterations in FAD binding (Figure 6), particularly the mutation T16M that affects a residue in close contact with the FAD molecule.

Inspection of those regions of the protein that show high stability and/or strong ligand binding-dependent stability may provide further explananation for some of the functional alterations displayed by mutant NQO1 proteins. In Figure 9A–D, we show the highly stable regions (with minimal HDX after 3 h) in NQO1_apo_, NQO1_holo_ and NQO1_dic_ (see also Appendix A). Residues L7 and V9 belong to this stable core in any of the three ligation states, whereas G3 does not in any of these ligation states (Figure 9A–D). The residue A29 is adjacent to this core in all three ligation states. In Figure 9E,F, we show those protein regions that undergo large stability changes upon binding FAD (Figure 9E) and Dic (Figure 9F). Regarding FAD binding, residues T16 and Y20 are buried in these regions that undergo large stability changes upon binding, thus supporting that their effects might be associated with alterations of the structural stabilization provided by FAD binding. A similar interpretation can be proposed to explain the effects of the mutation A29T, which affects a residue adjacent to those regions largely stabilized by FAD binding. Consequently, the lower affinity for FAD in the mutants T16M, Y20N and A29T could be partially explained by their perturbing effects on regions stabilized upon FAD binding (Figure 9E). A similar interpretation could apply for the lack of effect of the mutations studied in this work on Dic binding, since overall, the residues affected by these mutations are located in regions that are not largely stabilized upon inhibitor binding (Figure 9F).

### 3.8. Correlations between Loss-of-Function Scores Derived from Experimental Analysis of Mutational Effects and Bioinformatic Tools

So far, we have shown that naturally-occurring mutations, found in either targeted (i.e., disease-associated) samples (COSMIC database) or whole-genome sequencing initiatives (gnomAD database), can affect, to different extents, diverse features in a multi-functional protein, such as NQO1. These results (compiled in the so-called *N_t_ set*) extend our previous knowledge on a variety of natural (*cancer-associated*, P187S and K240Q) as well as non-natural mutations generated at the P187 and K240 sites (named as *P187 and K240 sets*; see Table 2) [19,39]. To what extent can we predict the potential molecular and functional consequences (plausibly linked to their *potential pathogenicity*) of this set of 22 mutations using a combination of some of the most common bioinformatic algorithms (Table 4)? This task is challenging, since current algorithms usually provide a single metric to assess this potential pathogenicity, whereas, as we have shown here, mutational effects on protein functionality are intrinsically complex (i.e., the protein is multi-functional). In an attempt to assess our capacity to establish correlations between experimental analysis and bioinformatic predictive tools, we have determined a simple experimental score (ES) for this set of 22 mutations that takes into account, in a semi-quantitative manner, their effects on different functional features (Table 2). Although these are features characterized in vitro, we may expect that some of them can reflect loss-of-function phenotypes in vivo (and, thus, potential pathogenicity due to loss-of-function). For instance, severe folding/solubility problems, impaired thermal stability and reduced local stability are likely associated with folding and stability problems inside cells (e.g., the consequences of P187S on these features are clearly associated with its intracellular stability; [41,43,52,68]). In addition, reduced local stability and FAD binding affinity can be also associated with low intracellular activity and stability due to increased population of the inactive and degradation-prone apo-state (e.g., in the phosphomimetic mutant S82D; [91]). Reduced affinity for the inhibitor Dic may also reflect conformational alterations associated with enhanced intracellular degradation (the P187S polymorphism; [41]) and/or reduced intracellular sensitivity towards NQO1 inhibition, leading to reduced cancer growth [39,44,63,93].

ESs were meant to provide a semiquantitative ranking of pathogeniticy: an ES of 1 corresponded to a *severe* mutation, ES of 2 to a *mild* one and ES of 3 to a *neutral* mutation. Calculation of the ESs for this set of mutants showed that five mutations caused large defects on protein stability and/or solubility, and thus, these were considered as severe (i.e., ES of 1), including the mutations L7P and L7R (Table 2). The remaining 17 mutations showed an ES ranging from 1.4–3.0. Among the lowest ESs, we found the polymorphism P187S, the *gold standard* of loss-of-function in NQO1 [41,43,52,68]. Most of the mutations in the N_t_ and K240 sets showed mild to neutral ESs (ES from 2.2 to 3), whereas variants in the P187 set were more deleterious (ES from 1 to 2).

We used these ESs to ascertain whether popular bioinformatic tools would predict and rank potential pathogenic effects. To this end, we generated bioinformatic scores (BSs) to be semiquantitatively compared with the ESs (Table 3 and Table 4). The results from these correlations are shown in Figure 10. When the full mutant set was analyzed, we observed a significant positive correlation between ESs and BSs (Figure 10). Visual inspection of this plot shows this correlation performs differently for different sets of mutations. The correlation for the N_t_ set is much stronger than that for the P187 set and, particularly, than that for the K240 set (Figure 10). The origin of this difference in performance is unclear. However, the very poor perfomance with the K240 set might have a straightforward explanation: this residue is highly conserved among mammalian NQO1 sequences, thus likely explaining that bioinformatic tools may identify mutations in this residue as potentially pathogenic. However, experimental analyses have shown that even highly disrupting mutants (such as K240E and K240G) still show many functional aspects not departing much from those of WT NQO1 [19,39]. These results were somehow expected: current bioinformatic approaches may provide some good overall results for genotype–phenotype correlations but fail to predict accurately and individually this correlation for individual mutations (one of the holy grails of personalized medicine, [99]), particularly when mutations affect highly conserved residues.

## 4. Conclusions

From the experimental characterization of the effects of eight naturally-occurring mutations in the multifunctional NQO1 protein, combined with detailed structural analysis and bioinformatic predictions, we can draw several important conclusions and propose future research lines.

First, by investigating the consequences of missense mutations found in whole-genome sequencing studies (with different allelic frequencies; see Table 1) as well as those found in cancer cell lines, we have observed that the severity of the pathogenic effects does not correlate well with their presence in cancer samples (i.e., the mutations L7P and L7R cause similarly catastrophic effects) nor with frequency in untargeted whole-genome sequencing studies (e.g., L7R, a severe mutation, has a 6.5-fold higher allelic frequency than G3S, a neutral mutation). Interestingly, one of the most deleterious natural missense variants in NQO1 (P187S) is also the most common in the human population (with an allelic frequency over 0.3) and is the missense variant most robustly associated with predisposition to disease [47,65]. These results may apparently contradict recent large-scale mutational analysis in several disease-associated protein systems that showed a negative correlation between allelic frequency of missense mutations and their effects on structural and interacellular stability [5,9,10,26]. This might imply that the correlation between allelic frequency and potential pathogenicity is not universal, that NQO1 is just an odd exception or that incorporating multiple functional features (stability, activity, regulation, …) into the picture may lead to more complex correlations. Obviously, understanding these and other possible scenarios deserves further research, including the analysis of larger sets of naturally-occuring missense mutations in diverse disease-associated proteins.

Second, structural–energetic–functional relationships of mutational effects are complex, and thus, an integrated view from biochemical and biophysical functional and stability studies, structural modeling and energetic considerations provides a suitable approach to characterize mutational effects in multifunctional proteins. Ideally, this should be also complemented with expression studies and stability analyses in eukaryotic cells. Obviously, all of this cannot be done at a whole-genome scale. However, at a smaller scale, these studies could provide insight into key concepts to be implemented in current *in silico* approaches for large-scale genotype–phenotype predictions. For instance, it is particularly relevant to consider the locally destabilizing effects of mutations and how these effects can propagate to distant sites in protein structures, thus contributing to pleiotropic (i.e., mulfunctional) effects of missense mutations [19,22,27,28,29,39,100]. It is also of interest that the threshold to cause intracellular destabilization with pathogenic consequences seems to be quite low, in the order of 2–3 kcal·mol^−1^ (often determined as the effects on overall conformational stability from experimental or computational methods) [9,10,26]. We should also consider local stability effects, not only for mutational effects on intracellular stability but also for functional consequences. Although P187S is known to cause devastating effects on intracellular stability and function, these effects are associated with changes in local stability in the order of 2–3 kcal·mol^−1^ [41], and effects within this range are observed for other naturally-occurring variants (such as T16M, Y20N, A29T and K240Q) (Table 3). Since propagation of mutations effects to distant sites in protein structure is likely a universal behavior [31,101,102,103], we would anticipate that the pleitropic effects described for missense mutations in NQO1 could apply to many other disease-associated protein systems [19].

Third, it is also of interest that scores based on quantitative analysis of mutational effects and those built using diverse current bioinformatic tools provided a positive correlation for a set of 22 mutations on NQO1. However, as we have shown, this predictive power is still disappointing when genotype–phenotype correlations are attempted for single mutations and also performs quite poorly for certain mutated sites (e.g., the K240 site). Thus, our work further supports the notion that detailed characterization of mutational effects by extensive functional and stability analysis will help to improve our capacity for accurate and large-scale genotype–phenotype correlations.

## Figures and Tables

**Figure 1 jpm-10-00207-f001:**
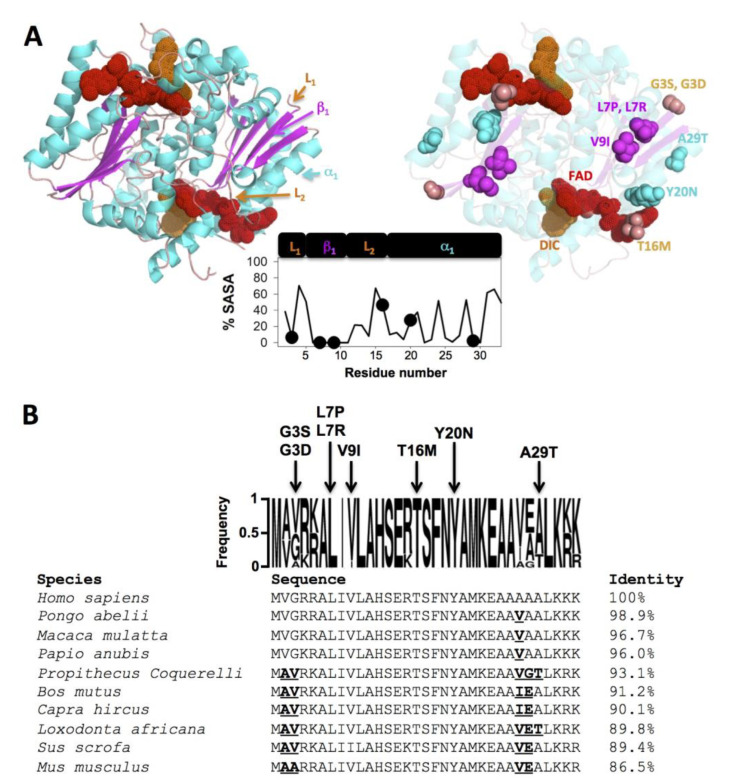
Structural location and sequence conservation of the NQO1 mutations studied in this work. (**A**) Structural representation on the dimeric structure of NQO1 (PDB code 2F1O; [72]). The left panel shows the secondary structure elements to which mutated residues belong (L_1_, residues 1–4; β_1_, residues 5–10; L_2_, residues 11–16 and α_1_, residues 17–33). The right panel shows the location of the mutated residues as well as the FAD and dicoumarol (Dic) molecules. Residues are colored according to the secondary structure. The plot in the middle shows the solvent accessibility (% SASA) of this region (determined using GETAREA on the PDB code 2F1O as the average of the two monomers in the dimer). (**B**) Sequence alignment of 10 selected NQO1 mammalian proteins. Residues in bold-underlined indicate non-conservative mutations vs. the human sequence (note that K-to-R or V-to-I are considered as *conservative*). The frequency plot over the NQO1 alignment (generated using WebLogo, https://weblogo.berkeley.edu/) also shows the identity of the missense mutations investigated in this work.

**Figure 2 jpm-10-00207-f002:**
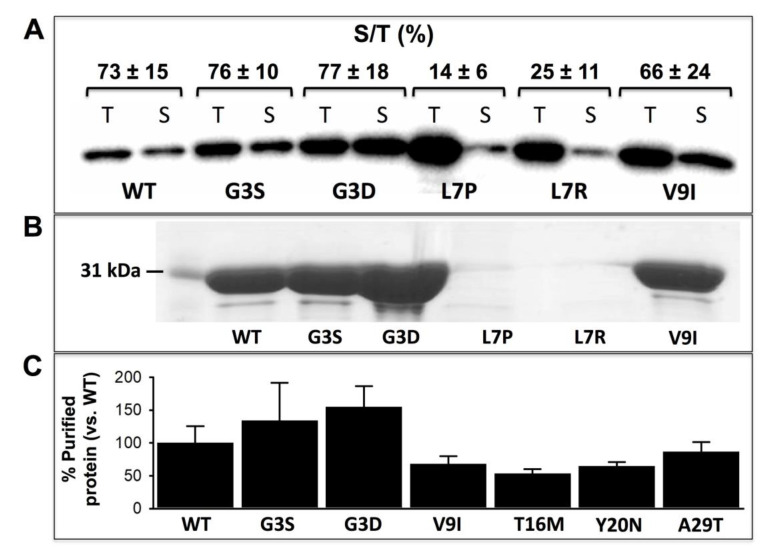
Expression and solubility of NQO1 variants in *E. coli* cells. (**A**) Western-blot analysis of NQO1 content upon expression in *E. coli* cells. Cells were sonicated to obtain total extracts (T), while soluble extracts (S) were obtained upon centrifugation at 21,000× *g* for 20 min at 4 °C. Samples were denatured with Laemmli buffer and submitted for Western-blot analysis using anti-NQO1 antibody, F-8 antibody (Santa Cruz Biotechnology). The Western-blot is representative from three different purifications. The fraction of the total protein found in the soluble extract upon densitometric analysis (S/T, as %) is indicated (as mean ± s.d. from three independent experiments). (**B**) Purified NQO1 proteins from three different purifications were concentrated ~10-fold and samples analyzed by SDS-PAGE. Note that during purification and storage, the remaining soluble protein of L7P and L7R was negligible. (**C**) Yield in NQO1 protein variants after immobilized-metal affinity chromatography (IMAC) purification. Data were the mean ± s.d. from 3–4 different purifications for each NQO1 variant. Wild-type (WT) levels were 1.45 ± 0.38 mg·L^−1^ of culture and used to normalize yields.

**Figure 3 jpm-10-00207-f003:**
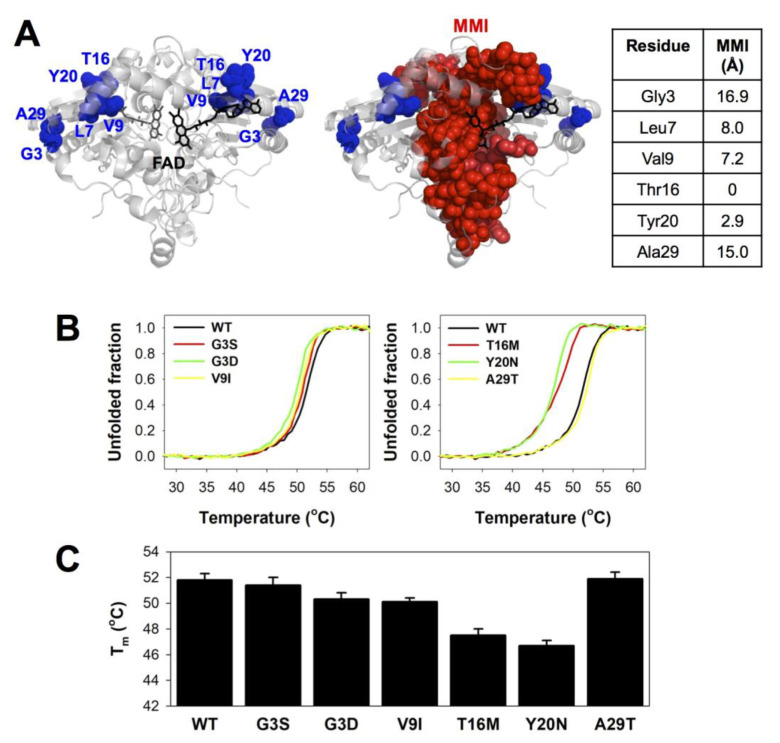
Thermal stability of NQO1 variants. (**A**) Structural location of the mutated residues, showing their proximity (as a minimal distance) to the monomer:monomer interface (MMI). The structure used for display has the PDB code 2F1O [72]. Residues belonging to the MMI were identified as described [90]. (**B**) Thermal denaturation profiles; (**C**) T_m_ values (mean ± s.d. from six replicas, using proteins from two purifications).

**Figure 4 jpm-10-00207-f004:**
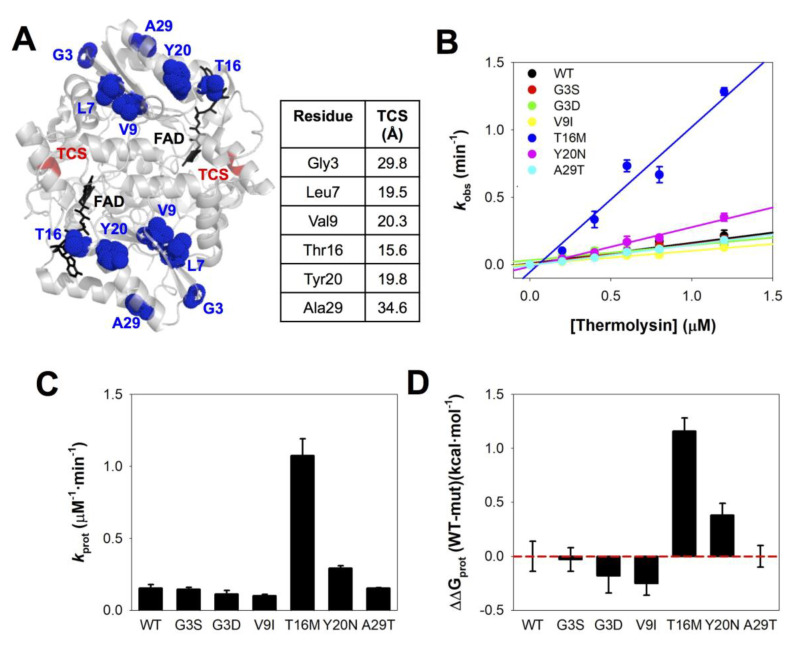
Local stability of the N-terminal domain (NTD) probed by proteolysis with thermolysin. (**A**) Location of mutated residues regarding the thermolysin cleavage site (TCS). The minimal distance between the mutated residue and the TCS backbone (Ser72-Val73) is also indicated (using the structure with PDB code 2F1O). (**B**) Linear dependence of apparent rate constants (*k*_obs_) on protease concentration. The slopes provide the values of the second-order rate constants *k*_prot_. (**C**,**D**) Values of *k*_prot_ for the NQO1 variants (**C**) and the effect of mutations on the local stability (**D**) as ΔΔG_prot_. Note that a positive value of ΔΔG_prot_ indicates a destabilizing effect.

**Figure 5 jpm-10-00207-f005:**
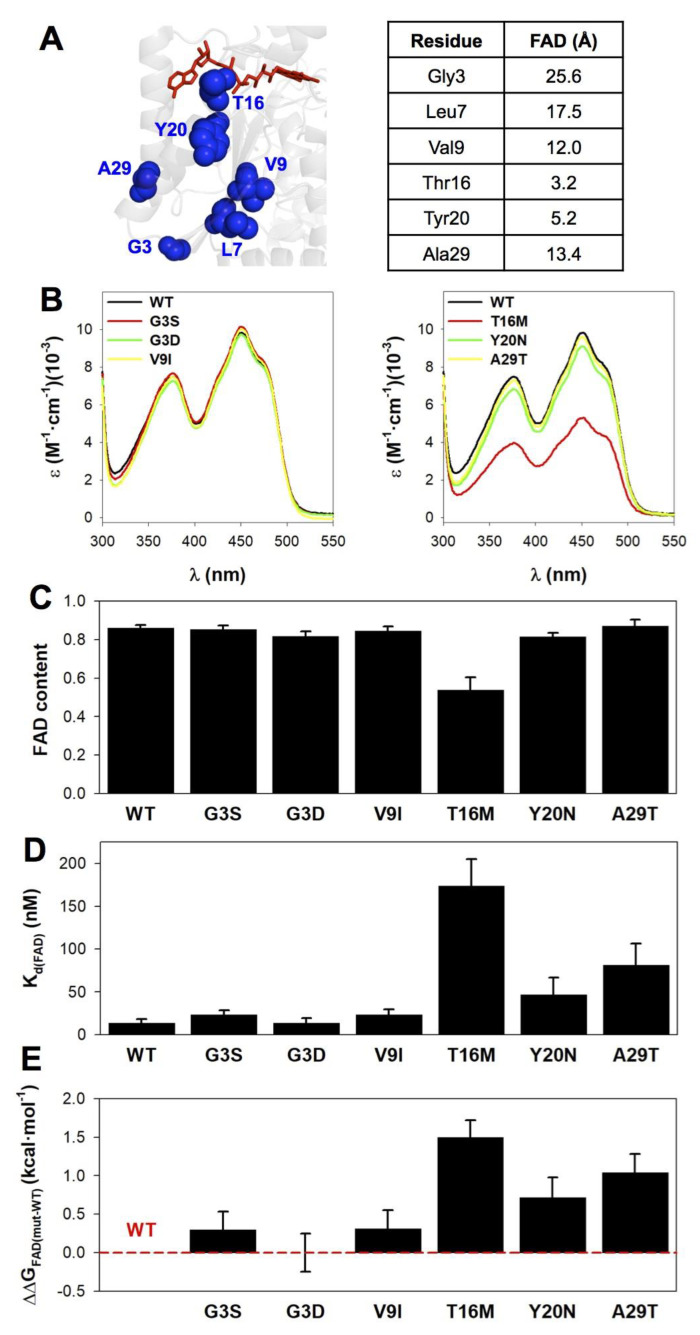
FAD content and binding affinity in purified NQO1 variants. (**A**) Location of mutated residues and minimal distances to the FAD molecule (using the structure with PDB code 2F1O). (**B**) Near-UV/visible absorption spectra of NQO1 variants. (**C**) FAD content (per NQO1 monomer) derived from absorption spectra (considering a ε_450_ = 11,300 M^−1^·cm^−1^). (**D**) FAD-binding affinity of NQO1_apo_ proteins determined by fluorescence titrations. (**E**) Difference in binding free energies (ΔΔG_FAD_) between a given mutant and the WT protein. Errors in ΔΔG_FAD_ are those determined from linear propagation. Data in B–C are mean ± s.d. from at least three different purifications, and in D–E, from two different titrations with two different preparations of apo-proteins.

**Figure 6 jpm-10-00207-f006:**
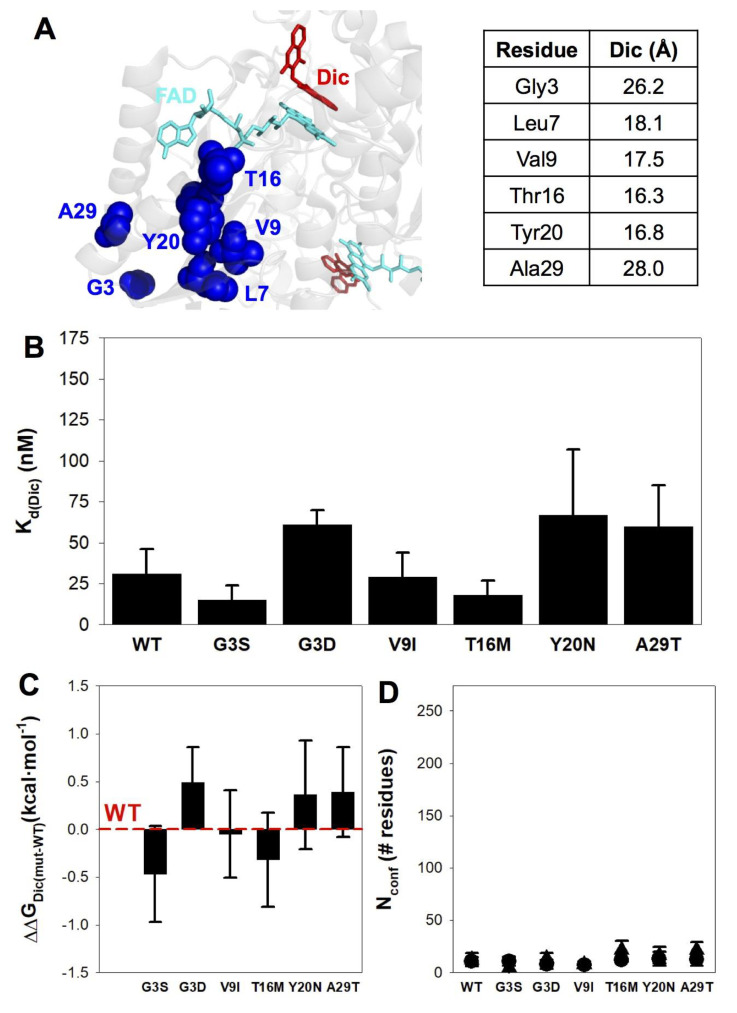
Affinity and structure–energetics analysis for Dic binding to NQO1 variants. (**A**) Location of mutated residues and minimal distances to Dic (using PDB code 2F1O). (**B**) Dissociation binding constants at 25 °C. Data are mean ± s.d. from at least three independent experiments for each variant. (**C**) Difference in binding free energy between a given mutant and the WT protein. Errors are those determined from linear propagation. (**D**) Magnitude of the conformational change (as a number of residues, N_conf_) determined from experimental binding enthalpies (circles) and changes in heat capacity (triangles).

**Figure 7 jpm-10-00207-f007:**
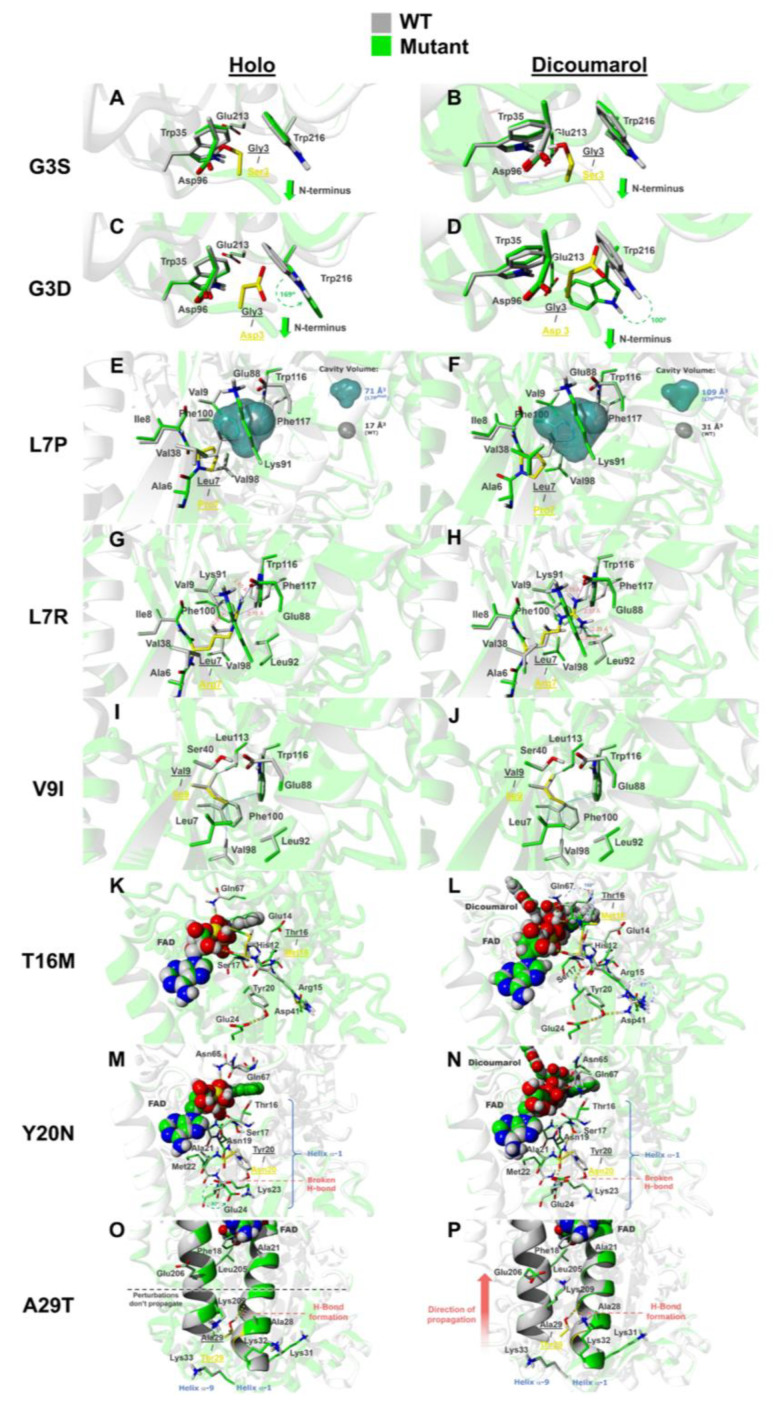
Overview of the mutational effects from in silico mutagenesis. Gray-colored amino acids represent the WT structure while green-colored amino acids represent the mutated models. Mutated residues are shown as sticks colored in yellow for each case. (**A**,**B**) NQO1_holo_ (**A**) and NQO1_dic_ (**B**) states with the G3S mutation. Green arrows indicate displacement of the N-terminus on the opposite side of the mutation. (**C**,**D**) NQO1_holo_ (**C**) and NQO1_dic_ (**D**) states with the G3D mutation. Green dashed arrows indicate side chain rotation. (**E**,**F**) NQO1_holo_ (**E**) and NQO1_dic_ (**F**) with the L7P mutation. Blue and gray surfaces represent cavities in the protein core for the mutant and WT, respectively. (**G**,**H**) NQO1_holo_ (**G**) and NQO1_dic_ (**H**) with the L7R mutation. Clashes are shown as blue solid lines marked with inter-atomic distances. (**I**,**J**) NQO1_holo_ (**I**) and NQO1_dic_ (**J**) with the V9I mutation. Hydrophobic interactions are shown as solid lines between interacting atoms colored gray and blue for the interactions created by V9 (WT) and I9 (V9I mutant) residues, respectively. (**K**,**L**) NQO1_holo_ (**K**) and NQO1_dic_ (**L**) with the T16M mutation. Ligand atoms are displayed in ball representation. Hydrogen bonds are depicted as yellow dashed cylinders. (**M**,**N**) NQO1_holo_ (**M**) and NQO1_dic_ (**N**) with the Y20N mutation. (**O**,**P**) NQO1_holo_ (**O**) and NQO1_dic_ (**P**) with the A29T mutation. Helices α1 and α9 are shown in ribbon representation.

**Figure 8 jpm-10-00207-f008:**
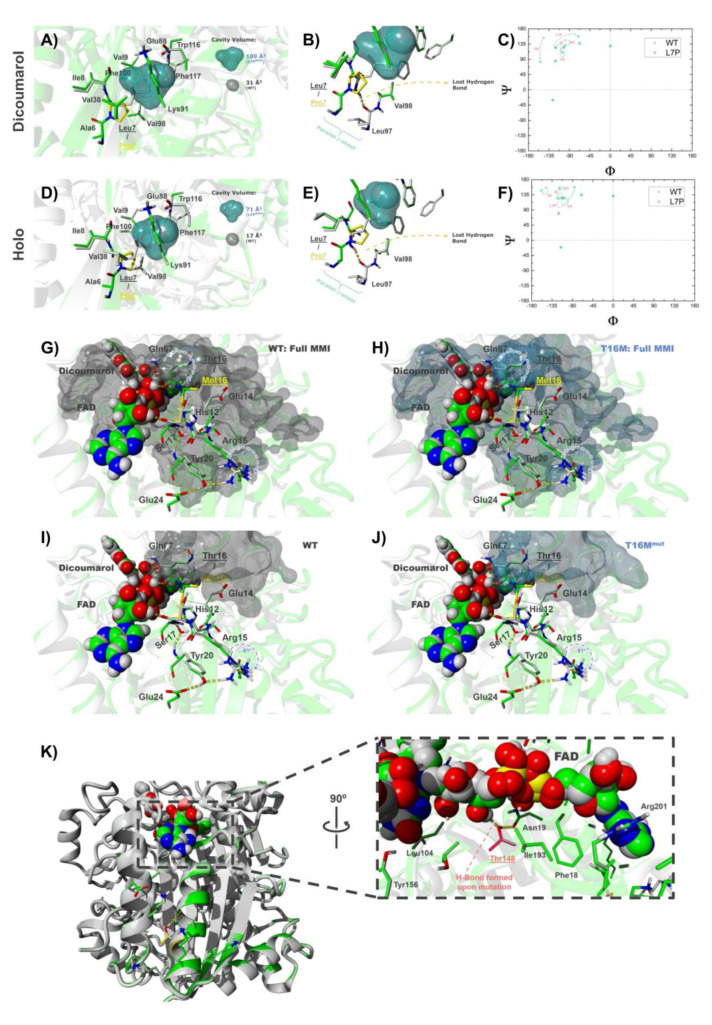
Detailed structural analysis of the mutations L7P (**A**–**F**), T16M (**G**–**J**) and A29T (**K**). (**A**) Overall view of the L7P site in the NQO1_dic_ state (mutated residue in yellow). The mutation generates a cavity in the hydrophobic core of the protein (semi-transparent blue for L7P and gray for WT protein). (**B**) Close-up view on the hydrogen bond disrupted by L7P in the NQO1_dic_ state. (**C**) Ramachandran plot of the first ten amino acids in the NQO1_dic_ state. Arrows indicate the displacement in dihedral angles caused by L7P mutation that affects the β-sheet pattern near the mutated site. (**D**) Close-up of the mutated site (in yellow) in the NQO1_holo_ state. The mutation L7P generates a cavity in the hydrophobic core of the protein represented in semi-transparent blue (L7P) and gray (WT). (**E**) Zoom-in on the hydrogen bond formerly established by L7 and disrupted by the L7P mutation in NQO1_holo_. (**F**) Ramachandran plot of the first ten amino acids in NQO1_holo_ showing, with arrows, the displacement caused by L7P mutation that affects the β-sheet pattern near the mutated site. (**G**–**J**) Effects of the T16M mutation on the MMI of the NQO1_dic_ state. The MMI surface is shown in gray for the WT and in bluish for the T16M mutant. The overall view is similar to that shown in Figure 7L. The complete MMI for the WT protein (**G**) vs. T16M mutant (**H**) is shown. Mutational effects are found in the top right corner. (**I**,**J**) Similar representations to those in panels (**G**,**H**) but only showing residues of the MMI close (<5 Å) to the mutated site. (**K**) Close-up view of the hydrogen bond formed between FAD and residue T148 in the A29T mutant in the NQO1_dic_ state.

**Figure 9 jpm-10-00207-f009:**
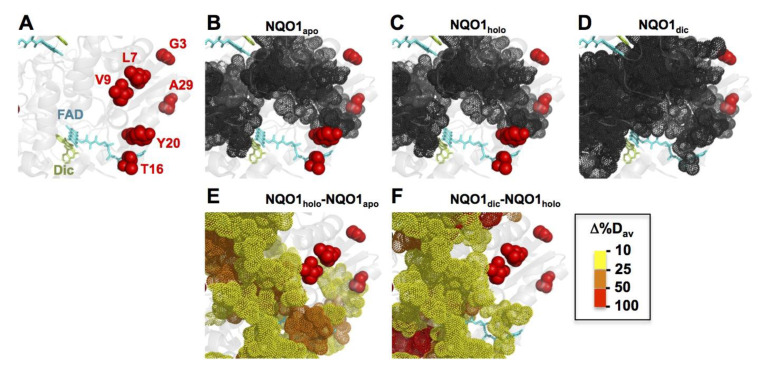
Location of mutated residues regarding local stability measurements by hydrogen/deuterium exchange (HDX). (**A**) Structural location of mutated residues regarding bound FAD and Dic. (**B**–**D**) Residues in dark grey/dot representation indicate those regions of the WT protein (NQO1_apo_, NQO1_holo_ and NQO1_dic_) that are highly stable (<20% HDX after 3 h). (**E**,**F**) Residues in dot representation (see color scale) indicate those whose stability (as Δ%D_av_) was increased between the two states (NQO1_holo_ vs. NQO1_apo_ and NQO1_dic_ vs. NQO1_holo_). Plots are generated using primary data from [90].

**Figure 10 jpm-10-00207-f010:**
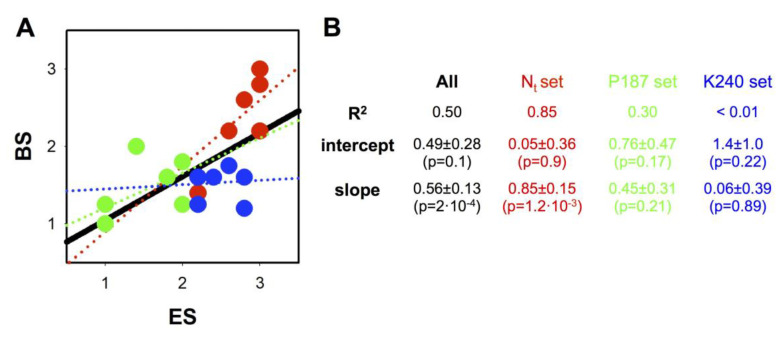
Correlation between experimental scores (ESs) and bioinformatic scores (BSs). (**A**) Plot of ES vs. BS for all mutants compiled in Table 3 and Table 4. Red circles show mutants belonging to the *Nt set* (the eight mutants characterized in this work) and green and blue circles show mutants belonging to mutations at the P187 and K240 sites, respectively (*P187* and *K240 sets*, see Table 3 and Table 4 [39]). The thick black line is the linear regression of all mutants, while dotted lines are linear fittings using one of the three sets (N_t_, P187 or K240 sets; red, green and blue lines, respectively). (**B**) Best-fit parameters for linear regressions are shown in panel B.

**Table 1 jpm-10-00207-t001:** NQO1 mutations experimentally characterized in this work.

Trivial Name	Amino Acid Change	Nucleotide Change	COSMIC	gnomAD V.2.1.1			ΔΔG (kcal·mol^−1^) ^1^
				All Samples (AS)	Non-Cancer (NC)	Ratio (AS:NC)	
G3S	p.G3S	c.7G > A	-	1.2 × 10^−5^	1.3 × 10^−5^	0.9	−1.5 ± 1.0
G3D	p.G3D	c.8G > A	+	-	-	-	−1.4 ± 0.9
L7P	p.L7P	c.20T > C	+	-	-	-	−4.3 ± 1.2
L7R	p.L7R	c.20T > G	-	7.8 × 10^−5^	7.5 × 10^−5^	1.0	−2.1 ± 0.8
V9I	p.V9I	c.25G > A	-	3.9 × 10^−5^	4.1 × 10^−5^	0.9	−0.4 ± 0.6
T16M	p.T16M	c.47C > T	-	2.8 × 10^−5^	2.6 × 10^−5^	1.1	−0.1 ± 0.2
Y20N	p.Y20N	c.58T > A	-	2.1 × 10^−5^	1.9 × 10^−5^	1.1	−1.1 ± 0.6
A29T	p.A29T	c.85G > A	+	-	-	-	−1.3 ± 1.3

^1^ ΔΔG is the average ±s.d. from the stability effects provided by Site Directed Mutator (SDM), PopMuSiC, DynaMut, MAESTRO and CUPSAT (see Appendix A).

**Table 2 jpm-10-00207-t002:** Mutational effects on conformational stability (as folding free energy between the mutant and WT proteins, ΔΔG, in kcal·mol^−1^, in the NQO1_holo_ and NQO1_dic_ states, determined by Rosseta). Negative values indicate a destabilizing effect whereas positive values indicate a stabilizing effect.

	G3S	G3D	L7P	L7R	V9I	T16M	Y20N	A29T
*NQO1_holo_*	−5.0	−11.6	−29.4	−14.1	1.9	−0.8	−8.1	−10.9
*NQO1_dic_*	−1.5	−10.1	−25.5	−35.9	0.6	0.8	−5.4	0.4

**Table 3 jpm-10-00207-t003:** Classification of phenotypic consequences due to NQO1 mutations based on experimental data. The experimental scores (ES) are calculated as the average from individual scores determined for each phenotypic trait as follows: *Expression:* +++, 50–100% of WT levels; ++, 20–50% of WT levels; +, <20% of WT levels; *Thermal stability:* +++, within 2 °C of WT T_m_; ++, within 2–5 °C of WT T_m_; +, >5 °C lower than WT T_m_; *Proteolysis:* +++, rate constant within three-fold vs. WT (0.65 kcal·mol^−1^); ++, within 3 to 30-fold vs. WT (2 kcal·mol^−1^); ++ >30-fold faster than WT (over 2 kcal·mol^−1^); *FAD binding:* +++, within three-fold of WT *K*_d_; ++, within 3–10 fold higher than WT *K*_d_; +, >10-fold higher than WT *K*_d_; *Dic binding:* +++, within three-fold of WT *K*_d_; ++, within 3–10 fold higher than WT *K*_d_; +, >10-fold higher than WT *K*_d_.

Variant	Expression	Thermal Stability	Proteolysis	FAD Binding	Dic Binding	ES
G3S	+++	+++	+++	+++	+++	3
G3D	+++	+++	+++	+++	+++	3
L7P	+					1
L7R	+					1
V9I	+++	+++	+++	+++	+++	3
T16M	+++	++	++	+	+++	2.2
Y20N	+++	++	+++	++	+++	2.6
A29T	+++	+++	+++	++	+++	2.8
P187S	+++	+	+	+	+	1.4
P187E	+					1
P187R	+					1
P187L	+					1
P187A	+++	++	+	++	++	2
P187G	++	+	++	+++	++	2
P187T	+++	++	+	++	+	1.8
K240Q	+++	+++	++	++	++	2.4
K240I	+++	+++	++	+++	+++	2.8
K240E	+++	+++	+	++	++	2.2
K240T	+++	+++	+++	++	+++	2.8
K240H	+++	+++	++	++	+++	2.6
K240A	+++	+++	+	+++	+++	2.6
K240G	+++	+++	+	++	++	2.2

**Table 4 jpm-10-00207-t004:** Classification of phenotypic consequences of NQO1 mutations based on bioinformatic analysis. The bioinformatic scores (BSs) are calculated as the average from individual scores provided by different algorithms as follows: *Polyphen-2:* +++, Benign; ++, Possibly damaging; +, Probably damaging; *SIFT:* +++, tolerated; +, Not tolerated; *Mutation Taster:* +++, polymorphism; +, disease-causing; *Provean:* +++, neutral; +, deleterious; PON-P2: +++, neutral; ++, unknown; +, pathogenic. * N.Det.- not determined because two nucleotide have to change.

Variant	PolyPhen-2	SIFT	Mutation Taster	Provean	PON-P2	BS
G3S	+++	+++	+++	+++	+++	3
G3D	+++	+++	+++	+++	++	2.8
L7P	+	+	+	+	+	1
L7R	+	+	+	+	+	1
V9I	+++	++	+	+++	++	2.2
T16M	+	+	+	+++	+	1.4
Y20N	++	+++	+	+++	++	2.2
A29T	+++	+++	+	+++	+++	2.6
P187S	+++	+++	+++	+	++	2
P187E	++	+	N.Det. *	+	+	1.25
P187R	+	+	+	+	+	1
P187L	+	+	+	+	+	1
P187A	++	+++	+	+	++	1.8
P187G	++	+	N.Det. *	+	+	1.25
P187T	++	+++	+	+	+	1.6
K240Q	+	+	+	+++	++	1.6
K240I	+	+	+	+	++	1.2
K240E	+	+	+	+++	++	1.6
K240T	+	+	+	+++	++	1.6
K240H	+	+	N.Det. *	+++	++	1.75
K240A	+	+	N.Det. *	+++	++	1.75
K240G	+	+	N.Det. *	+	++	1.25

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
