# Peer review of "Naturally-Occurring Rare Mutations Cause Mild to Catastrophic Effects in the Multifunctional and Cancer-Associated NQO1 Protein"

_jpm, 2020, doi:10.3390/jpm10040207_

Round 1
Reviewer 1 Report
The manuscript describes the results of carefully planned experiments aimed to establish the in vitro effects of mutations at NQO1 protein. The manuscript is extremely well-written and presented.
Author Response
The manuscript describes the results of carefully planned experiments aimed to establish the in vitro effects of mutations at NQO1 protein. The manuscript is extremely well-written and presented.
We thank the reviewer for his/her very positive consideration of our work.
English language and style are fine/minor spell check required.
The manuscript has been carefully revised.
Reviewer 2 Report
In this manuscript, Pacheco-Garcia et al predicted the effect of 8 NQO1 mutations on the protein. This might be of importance. Nevertheless, the manuscript needs substantial revisions.
The manuscript, particularly the introduction, is extremely extensive with a lot of redundant information and excessive citations (including authors' self-citations). In general, the text should be shortened significantly.
The introduction should be better structured with a focus on information essential for understanding the NQO1 structure, function, and aims of the study. The first half of the introduction can be omitted. Table 1/Figure 1 should rather be part of the Methods section. The total number of NQO1 amino acids is missing in the text.
In the conclusion part, the authors suggested (based on the allelic frequency of the P187S variant in NQO1) that mutation frequencies are not negatively correlated with pathogenicity. Given the fact that the cancer risk associated with NQO1 mutation heterozygosity is unknown (low risk of cancer with RR<1.5 was inconsistently shown for P187S homozygotes), I do not think NQO1 is not a good model for such a general statement.
I would appreciate the merging of tables 3 and 4.
The typos should be corrected.
Author Response
In this manuscript, Pacheco-Garcia et al predicted the effect of 8 NQO1 mutations on the protein. This might be of importance. Nevertheless, the manuscript needs substantial revisions.
We thank the reviewer for his/her positive consideration of our work and constructive criticisms.
The manuscript, particularly the introduction, is extremely extensive with a lot of redundant information and excessive citations (including authors' self-citations). In general, the text should be shortened significantly.
According to reviewer´s 2 and 3 suggestions, the introduction has been shortened as well as the number of references.
The introduction should be better structured with a focus on information essential for understanding the NQO1 structure, function, and aims of the study. The first half of the introduction can be omitted.
According to reviewer´s 2 and 3 suggestions, we have shortened the Introduction, particularly the first part regarding genetic variability in human genome and current capability to predict genotype-phenotype relationships. However, we have not fully omitted this section. Respectfully, we consider that these issues iare extremely relevant for the aims and conclusions of the manuscript, and therefore, we have mantain this part of the introduction, although it has been shortened to about half of its original size in the revised manuscript.
Table 1/Figure 1 should rather be part of the Methods section.
This is certainly an interesting and reasonable alternative. However, we think that our option is also adequate, since the some basic properties of the mutations studied are INTRODUCED in the last paragraph of the introduction. Therefore, and respectfully, we have mantained Table 1/Figure 1 in the Introduction section.
The total number of NQO1 amino acids is missing in the text.
This now introduced in line 91: “each monomer has two domains:” has been change to “each monomer has 274 residues divided into two domains:”
In the conclusion part, the authors suggested (based on the allelic frequency of the P187S variant in NQO1) that mutation frequencies are not negatively correlated with pathogenicity. Given the fact that the cancer risk associated with NQO1 mutation heterozygosity is unknown (low risk of cancer with RR<1.5 was inconsistently shown for P187S homozygotes), I do not think NQO1 is not a good model for such a general statement.
Clearly, this in an important issue. We never meant to establish that the rare mutations investigated in this work ARE associated with cancer development or treatment. However, one of the main goals of Personalized Medicine is to predict alterations in protein function due to genetic variations, and thus, to establish correlations with disease development and treatment. Therefore, here “pathogenicity” (or more precisely “potential pathogenicity”, as we indicate now in the revised version of the manuscript) relates to the precise mutational effects on diverse protein functions. Therefore, we have further toned down statements regarding “pathogenicity” in Section 3.8 (Lines 1071, 1078-1079, 1088-1089) and Section 4 (Line 1183) to indicate that loss-of-function phenotypes in NQO1 may indicate “potential pathogenicity”.
I would appreciate the merging of tables 3 and 4.
Tables 3 and 4 are shown separately to clearly distinguish between analyses carried out to derive experimental and bioinformatic scores. In addition, each table is large and contains plenty of information. Merging them would likely lead to a huge table with too much information, thus more difficult to be interpreted by the reader. Thus, respectfully, we have not followed this recomendation to make presentation of these data easier and clearer to the reader.
The typos should be corrected.
The manuscript has been carefully checked and corrected.
Reviewer 3 Report
Based on recent advances in DNA sequencing technology, the authors used biophysical, structural and bioinformatics approaches to investigate the effect of eight rare mutations on the stability and functional features of the NQO1 enzyme. I think the strategy followed and the results may be of general scientific interest. The structure of the paper is clear and well organised, the tables and figure are easy to interpret.
I have only few comments:
Although some mutations in NQO1 may are associated to cancer, the analyzes reported by the authors do not provide data directly related to the topic. The authors may consider modify the title of the study, without mentioning cancer.
It would be appropriate to shorten the Introduction sections, in order to make the reading more smooth and pleasant.
The explanation of each bioinformatics tool in paragraph 2.4 is not necessary. This paragraph could be summarized.
In the results, it would be appropriate to perform a densitometry of the western-blot, in order to evaluate quantitatively the differences in NQO1 expression between wt and mutants.
Please, state whether significant residue changes are conservative or for all analyzed mutations.
Some revision of the English language is recommended, paying more attention to punctuation too.
Author Response
Based on recent advances in DNA sequencing technology, the authors used biophysical, structural and bioinformatics approaches to investigate the effect of eight rare mutations on the stability and functional features of the NQO1 enzyme. I think the strategy followed and the results may be of general scientific interest. The structure of the paper is clear and well organised, the tables and figure are easy to interpret.
We are grateful for these positive comments.
I have only few comments:
Although some mutations in NQO1 may are associated to cancer, the analyzes reported by the authors do not provide data directly related to the topic. The authors may consider modify the title of the study, without mentioning cancer.
Certainly, the mutations investigated are rare and their association with cancer is not established. Even for P187S, a quite devastating polymorphism with a frequency of about 0.25, the association with cancer is often not statistically significant (particularly in heterozygosis). This is not unexpected since a WT copy of the gene would provide significant functionality in vivo (note that negative- dominant effects have not been described for NQO1 mutations). One of the main conclusions of this work is indeed reflected in the title: RARE mutations can devastate the function of NQO1, a PROTEIN ASSOCIATED WITH CANCER. Consequently, in our opinion, the title was not misleading. The association of NQO1 functions with cancer is well supported by scientific research over the last 25 years. We realize that NQO1 might not be a key player in cancer (in contrast to e.g. p53 or HIF1alpha). However, NQO1 functions involved in antioxidant defense, activation of chemotherapeutics and stabilization of key factors in cancer (actually p53 and HIF1alpha, among others), subtantiate its association with cancer development/treatment (always considering that cancer is a COMPLEX disease). Therefore, and respectully, we intend to mantain the title of the manuscript as it is.
It would be appropriate to shorten the Introduction sections, in order to make the reading more smooth and pleasant.
The introduction has been shortened accordingly.
The explanation of each bioinformatics tool in paragraph 2.4 is not necessary. This paragraph could be summarized.
This section has been largely summarized (see lines 440-471 in the revised manuscript).
In the results, it would be appropriate to perform a densitometry of the western-blot, in order to evaluate quantitatively the differences in NQO1 expression between wt and mutants.
We certainly agree with the reviewer. We have modified Figure 2 to include the quantitative analysis from densitometry, expressed as % of protein found in soluble/total extracts, as a simple metric of solubility. Methods and the legend of this figure has been modified accordingly to describe this change.
Please, state whether significant residue changes are conservative or for all analyzed mutations.
According to this suggestion we have added the following sentence in the introduction (lines 263-265 of the revised manuscript): “We may only consider the mutation V9I as conservative, since the rest of mutations will cause either large changes in backbone conformation/flexibility or side-chain size, polarity or charge.”
Some revision of the English language is recommended, paying more attention to punctuation too.
The manuscript has been carefully checked and corrected.